# Simulation Optimization for Complex Multi-Domain Physical Systems Based on Partial Resolving

**Kexi Hou and Yaohui Li ***

College of Mechanical and Electrical Engineering, Xuchang University, Xuchang 461000, China;
houkexi@yeah.net
* Correspondence: lyh@xcu.edu.cn

**Abstract:** The iterative process of simulation optimization is a time-consuming task, as it involves executing the main simulation program in order to evaluate the optimal constraints and objective functions repeatedly according to the values of tuner parameters. Parameter optimization for a model of a multi-domain physical system based on *Modelica* is a typical simulation optimization problem. Traditionally, each simulation during each iterative step needs resolve all the variables in all the mass differential-algebraic equations (DAE) generated from the simulation model through constructing and traversing the solving dependency graph of the model. In order to improve the efficiency of the simulation optimization process, a new method named partial simulation resolving algorithm based on the set of input parameters and output variables for complex simulation model was proposed. By using this algorithm, a minimum solving graph (MSG) of the simulation model was built according to the set of parameters, constraints, and objective functions of the optimization model. The simulation during the optimization iterative process needs only to resolve the variables on the MSG, and therefore this method could decrease the simulation time greatly during every iterative step of the optimization process. As an example, the parameter optimization on economy of fuel for a heavy truck was realized to demonstrate the efficiency of this solving strategy. This method has been implemented in MWorks—a *Modelica*-based simulation platform.

**Keywords:** simulation optimization; *Modelica*; multi-domain simulation; partial resolving; minimum solving graph; differential-algebraic equations

---

## 1. Introduction

### 1.1. Modeling and Simulation Technology for Multi-Domain Physical System Based on Modelica

Modeling languages for multi-domain physical systems have been developed since the 1970s [1]. In 1996, the uniform modeling language *Modelica* came into being, which inherits several excellent properties of previous modeling languages: object-orientation, non-casual modeling, multi-domain modeling capability, declarative modeling, and continuous-discrete hybrid modeling. *Modelica* is bent on serving as a standard language so that different domain models can be expressed and exchanged under a unified platform. Some powerful open libraries are now available from websites [2]. A great deal of basic models from multi-domains, including mechanical, electrical, hydraulic, thermodynamic, and control system are defined in these libraries. Based on these libraries, it is easy to build users' own models and to expand them through customizing models for reuse.

There are several applicable modeling and simulation tools based on *Modelica*, such as Dymola [3,4], Math *Modelica* [5,6], and Open *Modelica* [7,8]. Up until now, these are the three most influential and powerful software systems based on *Modelica*. In view of the development trend of *Modelica*, our group started with basic research on its theories and applications from 2002. After deeply studying the syntax

structure and system architecture of *Modelica*, we made further research on the solving strategy for large scale continuous-discrete hybrid differential-algebraic equations (DAE) [9], and exerted ourselves to implement a hybrid modeling platform based on *Modelica* for multi-domain physical systems [10,11]. Now, our work has come into being as a prototype software system—MWorks [12]—which provides six main modules: integrated modeler, compiler, analyzer, solver, optimizer, and postprocessor.

## 1.2. Simulation Solving for a Modelica Model

*Modelica* uses differential algebraic equations (DAE) to describe physical phenomena and behaviors of heterogeneous physical systems, and the topologies of physical systems are depicted by components and the connections among components. Therefore, the task of modeling and simulation based on *Modelica* is to construct and solve DAE [13]. The following three tasks are mainly used to solve a DAE system: symbolic simplifying, equation debugging, and numeric solving. Symbolic simplifying includes eliminating the equivalent equations [14], BLT (block lower triangular) partitioning [15], and reducing DAE index [16,17]. *Modelica* equation debugging [18] can determine whether a DAE equation system is a well-constrained system, an over-constrained, or an under-constrained system. Only a well-constrained system can be solved by using numeric solutions. After BLT-partitioning of the equation system, a series of sorted blocks are generated, including linear algebraic equations, non-linear algebraic equations, ordinary differential equations, DAEs, and hybrid DAEs, which could be resolved by calling DASSL (a differential algebraic system solver), Sundials, or other numeric solvers.

Decomposition of DAE systems can decrease the size of equations which should be solved simultaneously. Besides BLT partitioning, Secchi et al. [19] promoted the so-called waveform relaxation (WR) method, an operator-splitting approach to the solution of such DAE systems, which partitions the problem into several lower order subsystems. In addition, a novel partitioning algorithm [20] for the optimistic distributed simulation of hierarchical modular discrete event system specification (DEVS) models has been proposed.

Parallel processing is a potential and significant method to improve simulation efficiency. Combined the abstraction power of *Modelica* with support for shared memory bulk-synchronous parallel programming, flexible and simple *NestStepModelica* is proposed by Kessler et al. [21]. Lundvall and Fritzson [22] reported preliminary results of automatically generating parallel code from equation-based models together at two different levels. A unifying framework for specifying DEVS parallel and distributed simulation architectures [23] results in algorithms with a highly parallelizable concurrent fraction and low sequential overhead, which are especially suitable for coarse- and medium-grain MIMD (Multiple Instruction Stream Multiple Data Stream) distributed-memory machines.

## 1.3. Parameter Optimization for a Modelica Model

Parameter optimization for a *Modelica* model is a typical dynamic simulation optimization. It is highly time-consuming because simulations should be done to evaluate the value of constraints and objective functions during each iteration process.

For dynamic optimization, a control vector parameterization simulation solver is researched by Ko et al. [24]. Feehery and Bartona [25] described a new method with path constraints for solving dynamic optimization problems on the state variables. Huang et al. [26] presented a decomposition strategy, which combines the advantages of the control parameterization and has a simultaneous approach for solving dynamic optimization problem with path constraints. A decomposition strategy is developed by Logsdon and Biegler [27] to exploit the block matrix form of the discretized differential equations. It resulted from using collocation on finite elements and performed the optimization in the control space.

## 1.4. Method and Structure of This Work

Like Dymola and Math*Modelica*, MWorks can realize parameter optimization of a *Modelica* model by using the optimizer module. As we know, the iterative process of simulation optimization is

a computationally expensive task, as it requires executing the main simulation model and then evaluating the optimal constraints and objective functions repeatedly according to the values of tuner parameters. Parameter optimization for multi-domain physical systems under the simulation platform based on *Modelica* is a typical simulation optimization problem. Each simulation executed during the iterative process needs to resolve all the variables in the mass differential algebraic equations traditionally. In order to improve the efficiency of the simulation optimization process, we promoted a new method named the "partial resolving algorithm" for complex simulation models in MWorks. By using this algorithm, a minimum solving graph (MSG) of the simulation model according to the set of parameters, constraints, and objective functions of the optimization model was built. Therefore, simulation executing needs only resolve the variables in the MSG, and this method could decrease the simulation time greatly during each optimization iterative step. This paper will describe the new feature of MWorks to perform parameter optimization for *Modelica* simulation models, and generate in-depth research on the partial resolving algorithm and MSG generation during the optimization process for complex multi-domain physical models.

This paper is organized as follows: Section 2 introduces modeling and simulation processes for a *Modelica* model with a shunt-wound circuit example. Section 3 illustrates the design optimization process for a multi-domain system based on *Modelica*. Section 4 presents the concepts and corresponding algorithms of partial resolving strategy and then presents the minimum simulation solving algorithm according to the MSG. An application sample of our approach is demonstrated in Section 5.

## 2. Modeling and Simulation for a *Modelica* Model under MWorks

### 2.1. Modeling Example

MWorks modeler provides four views for modeling: FullText, Icon, Diagram, and HTML (Hyper Text Markup Language). FullText view is a text edit view to show and modify the mo text directly. Icon view is a drawing view to show and modify the graphical icon of the current model. Diagram view is another drawing view to show and modify the components and connections of the current model. HTML view is a view to show the HTML format content of the current model. As an example supported by the *Modelica* standard library, a parallel connected circuit model was built through drag-drop modeling under MWorks graphical modeling interface (Diagram view), as shown in Figure 1a. Its *Modelica* text can be updated automatically in the text modeling interface (FullText view), as shown in Figure 1b.

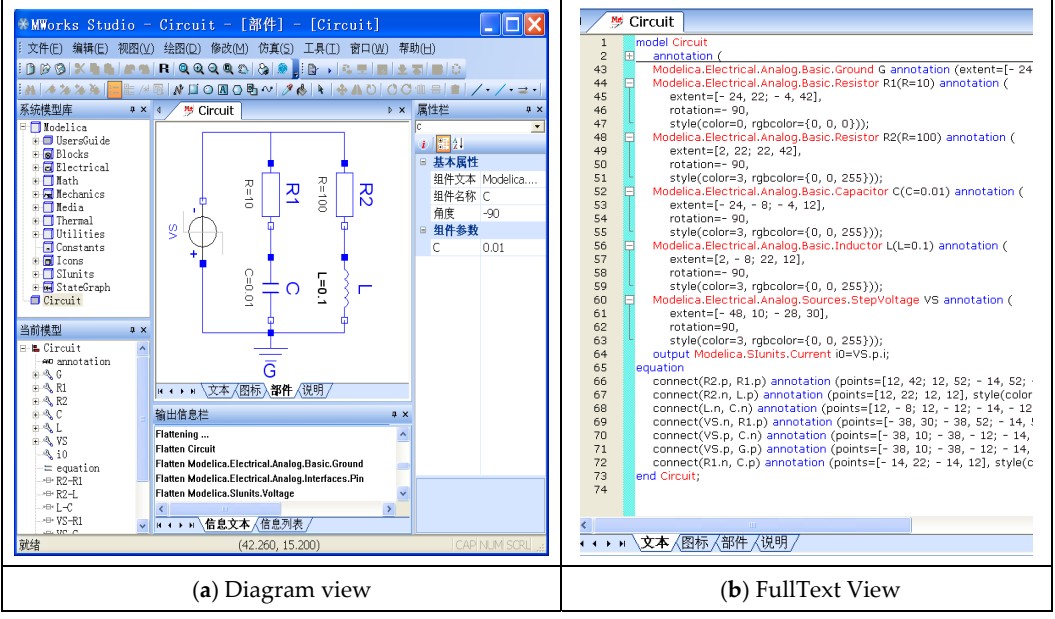

| (**a**) Diagram view | (**b**) FullText View |

**Figure 1.** Simple electrical circuit model.

Topology of a physical system such as the circuit model described above is depicted by components (instances of class) and connections between connectors of the components. Generally, equations are described by assigned expression, equivalent expression, or modifications. Equations can also be specified by *Modelica* connections. For example, a connection between two connectors *R1. p* (one node of resistor *R1*) and *C. p* (one node of capacitor *C*) implies two equations, namely *R1. p. v = C. p. v* and *R1.i + C. p. i = 0*. The former equation indicates that the voltages on the connected nodes are the same, and the latter corresponds to Kirchhoff's current law stating that the currents sum to zero at a node. The sum-to-zero equations are generated when the prefix *flow* is used. Similar laws apply to flow rates in a piping network and to forces and torques in mechanical systems [1].

In addition, it should be also noted that the keyword parameter in *Modelica* specifies a variable remaining constant during a simulation run, but can change values between runs. This makes it simple for users to modify the behavior of a model by setting different values to parameters, and the design optimization for a *Modelica* model is realized through tuning parameters to achieve minimum objective functions and to meet constraints.

## 2.2. Equations Generating and Sorting

Under the MWorks platform, the compiler translates the *Modelica* model into a flat DAE system firstly through semantic analysis; then the analyzer processes the DAE system into a series of sorted blocks [9], which can be solved in turn. As for the circuit model in Figure 1, the sorted 17 blocks are generated as follows:

⟨B1⟩ G.p.v = 0;

⟨B2⟩ C.n.v = G.p.v;

⟨B3⟩ L.n.v = C.n.v;

⟨B4⟩ VS.n.v = C.n.v;

⟨B5⟩ VS.signalSource.outPort.signal = VS.signalSource.p_height;

⟨B6⟩ VS.signalSource.y = VS.signalSource.outPort.signal;

⟨B7⟩ VS.v = VS.signalSource.outPort.signal;

⟨B8⟩ VS.v = VS.p.v-VS.n.v;

⟨B9⟩ R1.p.v = VS.p.v;

⟨B10⟩ R2.p.v = R1.p.v;

⟨B11⟩ R1.v  =  R1.p.v-R1.n.v;      0  =  R1.p.i  +  R1.n.i;      R1.i  =  R1.p.i;
      R1.R  *  R1.i  =  R1.v;      C.p.i  +  R1.n.i  =  0;      R1.n.v  =  C.p.v;
      0  =  C.p.i  +  C.n.i;      C.i  =  C.p.i;      C.i  =  C.C  *der(C.v);
      C.v = C.p.v-C.n.v;

⟨B12⟩ R2.v  =  R2.p.v-R2.n.v;      0  =  R2.p.i  +  R2.n.i;      R2.i  =  R2.p.i;
      R2.R*R2.i  =  R2.v;      L.v  =  L.p.v-L.n.v;      0  =  L.p.i  +  L.n.i;
      L.i  =  L.p.i;      L.L*der(L.i)  =  L.v;      L.p.i  +  R2.n.i  =  0;
      R2.n.v = L.p.v;

⟨B13⟩ R1.p.i + R2.p.i + VS.p.i = 0;

⟨B14⟩ VS.i = VS.p.i;

⟨B15⟩ 0 = VS.p.i + VS.n.i;

⟨B16⟩ C.n.i + G.p.i+L.n.i + VS.n.i = 0;

⟨B17⟩ i0 = VS.p.i;

Among the equations above, variables in italics represent parameters which are set before simulation; variables in bold mean that they will be solved in the block.

In addition, the specific meaning of each block from [B1] to [B17] can be clearly explained by Table 1.

**Table 1.** The meaning of B1–B17.

| Block Number | Meaning |
|---|---|
| B1 | Grounding voltage block |
| B2 | Capacitance voltage block |
| B3 | Inductance voltage block |
| B4 | Voltage source voltage block |
| B5 | Voltage source output signal block |
| B6 | Voltage source objective signal block |
| B7 | The relation block on Voltage source voltage and Voltage source output signal |
| B8 | Voltage source voltage relation block |
| B9 | Voltage block on Resistance 1 (*R1*) |
| B10 | Voltage block on Resistance 2 (*R2*) |
| B11 | The relation block on *R1*, Capacitance, Voltage, and Current |
| B12 | The relation block on *R2*, inductance, Voltage, and Current |
| B13 | The Current relation block on Resistance 1, Resistance 2, Voltage source |
| B14 | The Current block of Voltage source |
| B15 | The Current relation block of Voltage source |
| B16 | The Current relation block on Capacitance, Inductance, Grounding port, and Voltage source |
| B17 | Trunk current block |

### 2.3. Simulation

After generating the sorted equations, MWorks solver module makes the series of sorted blocks into a C code and executable program (.dll in MWorks), and then DASSL or Sundials solver is called to resolve all blocks in turn at each discretized time sequence. Finally, the results will be saved into a mat format file.

In general, full simulation is done when solving, i.e., all the variables in all the equations are resolved when simulating. The postprocessor can read the mat file and then draw the desired variable-time curves, and all these variables form a tree according to the hierarchical components. Figure 2 shows variable tree and the curves of three variables of the model in Figure 1—the total current *i0* and the two branch currents, R1.i and R2.i.

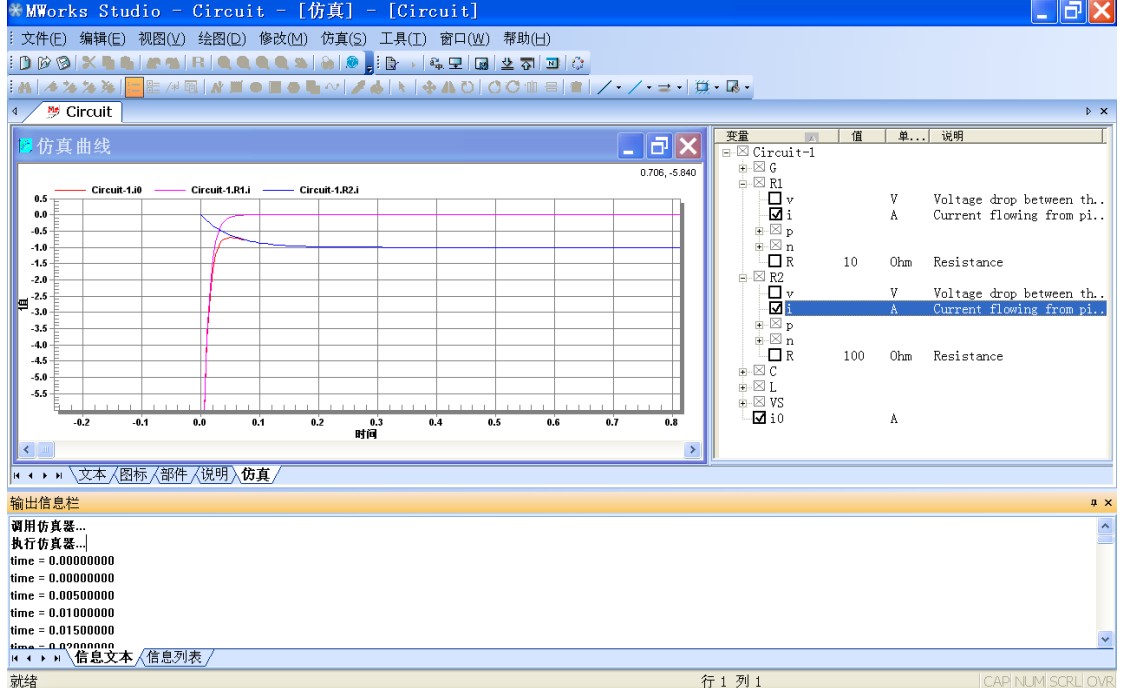

**Figure 2.** Simulation results of the circuit in Figure 1.

## 3. Parameter Optimization Process for a *Modelica* Model

### 3.1. Criteria of State Variables

Because optimization for a *Modelica* model is a kind of simulation optimization, the state variables in the objective functions or constraints have different values during the simulation time. Thus, we should set up the criteria of state variables when building the optimization model. Maximum value, minimum value, final value, initial value, integrated value and average value, are several common choices.

Aside from this, there are other three types of criteria [28]: over-shoot, settling-time, and rise-time. As shown in Figure 3a, the parameter final_Value is the desired reference value of the input signal y. The over-shoot is defined as max (0, y—Final_value), i.e., the largest value of y over the Final_value. Settling-time (Figure 3b) is the first time where the input signal y remains within a range of delta forever. Also, rise-time (Figure 3c) computes the rise time of the input signal. Generally, it is the time between the input signal being equal to 0.1 * Final_value and 0.9 * Final_value.

In addition, the difference (Figure 3d) between the ideal curve and the simulation response curve of a variable should be considered under the condition of parameter estimation. The ideal curve can be given with pairs of value-time series. The difference value can be calculated through interpolation. To make the difference of two curves minimize is to minimize the sum of all the squares of differences between ideal values and response values at the discrete time sequences.

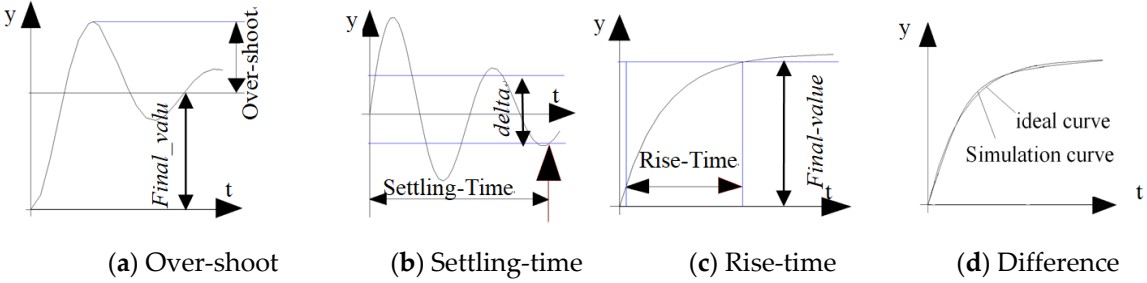

(**a**) Over-shoot　　　(**b**) Settling-time　　　(**c**) Rise-time　　　(**d**) Difference

**Figure 3.** Criteria of state variables.

### 3.2. Optimization Modeling for a Modelica Model with Multi-Cases

By using a suitable search algorithm to minimize some objective functions and satisfy some constraints, the Optimization based on *Modelica* is can find a set of optimal parameters of a *Modelica* model. The *Modelica* model is called a "main simulation model", which should be a *Modelica* class or *model* in which the number of variables is equal to the number of equations.

As we know, an optimization model has three aspects: design variables (or tuners), constraints, and objective functions. As for the simulation optimization for a *Modelica* model, generally tuners are the parameters of the main simulation model or parameters of its components of sub-components. A tuner has three main data members: default value, minimum, and maximum value. The default values of tuners are the initial values when optimizing; maximum and minimum values of tuners are boundary constraints of the optimization model.

In order to avoid evaluating values of expressions of user-defined design functions (constraints and objective functions), we can define them as some output variables in the model text. Then, values of design functions can be evaluated through these output variables, which can be resolved when simulating. Because the design functions often include state variables, the corresponding output variables should be set with different criteria, as mentioned in Section 3.1.

If we optimize a model that has a different typical work status or structures, the multi-case optimization model can be used [28]. Cases describe different operating conditions that are distinguished by different values of a set of parameters, namely, case parameters. Generally, case parameters are discrete parameters and are kept at a fixed value in a case. One case has one set of case

parameters. Multi-case optimization modeling means that when optimizing a model, all the objective functions and constraints of all the cases must be considered. Thus, the multi-objects optimization and multi-constraints optimization are two core parts in multi-case optimization.

Multi-objects of multi-cases can be aggregated into one object according to the weight of each object of each case. The sum of the absolute of each weighted object is a common aggregation method. Then, the multi-case optimization model can be described as follows:

$$
\begin{aligned}
find \quad & X \\
min \quad & \Sigma \left| f_{ij}/d_{ij} \right| && ij \in F \\
s.t. \quad & c_{ij} \leq d_{ij} && ij \in I \\
& c_{ij} = d_{ij} && ij \in E \\
& x^i_{min} \leq x^i \leq x^i_{max} && x^i \in X
\end{aligned}
\tag{1}
$$

where $X$ denotes tuners; $F$, $I$, and $E$ denote the set of objective functions, inequality constraints, and equality constraints in cases, respectively, i.e., $f_{ij}$ denotes the $i$th object in the $j$th case. In order to achieve the same scale of different objective functions, a fixed demand value ($d_{ij}$, $ij \in F$) is needed for each object in each case. A demand value has the same unit as its corresponding object, which denotes the designer's preferences. Moreover, $\Sigma \left| f_{ij}/d_{ij} \right|$ is the overall object function of multi-objects of multi-cases.

As for the circuit model of Figure 1, we consider the following optimization problem:

```
find:   {C.C, L.L}
min:    Difference (i0, Ij ), j = 1,2,3;  dj = 1;
s.t.     settling-time (i0) < Tj, j = 1,2,3
        over-shoot(i0) ≤ Aj, j = 1,2,3
cases:  {R1.R, R2.R} = {20, 100; 50, 50; 100, 20}
        {T1, T2, T3} = {2, 3, 4}
        {A1, A2, A3} = {0, 0, 0}
```

where *C.C* is the value of the capacitor *C*; *L.L* is the value of the inductance *L*; and *i0* is the overall current of the circuit. There are three cases in the model with different case parameters *R1.R* and *R2.R*. *I1*, *I2*, *I3* are three ideal current curves corresponding to the three cases, respectively, and they are given with a series of value-time pairs.

### 3.3. Parameter Optimization Process

As mentioned in the previous sections, a physical system described by *Modelica* can be defined as a parameterized simulation model. Therefore, design optimization can be employed to tune parameters such that the system behavior is improved. After setting up an optimization model for a *Modelica* simulation model, a search algorithm (such as SQP-sequential quadratic programming, Hookjeeves, GA-genetic algorithm, etc.) is used to constantly adjust design variables, execute the simulation program and evaluate the design functions in order to minimize the objective functions and meet the constraints. The process of design optimization based on *Modelica* model is illustrated in Figure 4.

As shown in Figure 4, the optimization model including design variables, case parameters, objective functions, and constraints is built from the *Modelica* simulation model. Before iteration of optimization, a full simulation program is generated. During each iteration, the simulation program is then called after tuning the values of design variables. If the model has multi cases, a simulation program would be called several times (equaling to the number of cases) with different sets of values to the case parameters during one iteration step. Assume the number of cases is *nCases*, and the number of the optimization iterations is *nIters*, then the simulation will run *nCases \* nIters* times during the whole optimizing process. After simulation, criteria of the state variables in the objective functions or constraints will be evaluated according to the simulation results. A more specific process may be referred to in [9].

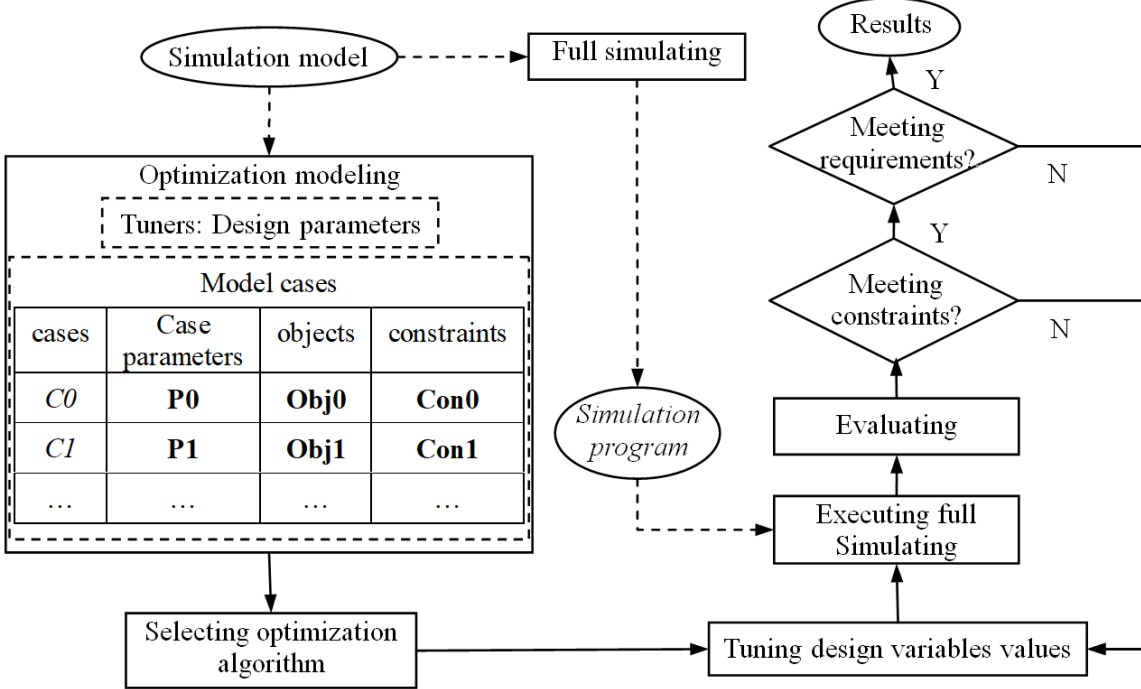

**Figure 4.** Process of design optimization for a *Modelica* model.

## 4. Partial Resolving Algorithm for a *Modelica* Model

As we know, many repetitive simulations must be executed during optimizing, especially for a multi-case optimization model. Thus, if the simulation program could be shortened, a great deal of time could be saved. Under MWorks, we adopt a method named "minimum simulation" in which full resolving is substituted with partial resolving.

The random geometric graphs [29] have been well-studied and applied to alternative mechanisms wrapped in real-world complex networks. In addition, the idea on the sorted blocks is interesting. It can potentially be used to structure insightful hierarchical structures. Shang [30] studied a model for inhomogeneous long-range percolation on the hierarchical lattice $\Omega_N$ of order $N$ with an ultrametric $d$. The random geometric graphs and the idea on the sorted blocks are helpful for researchers to further understand the partial resolving algorithm.

Note that this method can only be used under the condition that the equation system of the simulation model does not change when tuning the parameters. If the structure of the equation system changes, all the variables, the solving dependency graph, and the MSG could also change.

*4.1. Solving Dependency Graph*

**Definition 1.** *Coupling block, a set of tightly coupling simultaneous equations that can be solved independently. A coupling block has five domains:*

$$B = \{s\_sEqu, s\_sPar, s\_sVar, s\_pParents, s\_pChildren\},$$

*s_sEqu represents the set of equations; s_sPar represents the set of parameters; s_sVar represents the set of output variables which will be resolved through this block; s_pParents represents the set of pointers of parent blocks which should be resolved before; and s_pChildren represents the set of pointers of children blocks which depends on this block.*

**Pre-blocks** *of a block B is the blocks that are B's parents or their parents, recursively.*
**Post-blocks** *of a block B is the blocks that are B's children of their children, recursively.*

**Definition 2.** *Solving a dependency graph is a graph which is defined as follows:*

$$SDG = (\mathbf{B}, \bar{\mathbf{E}}).$$

*Vertices **B** in the solving dependency graph (SDG) is a set of coupling blocks and the edge$(B_1, B_2) \in \bar{E}$ indicates the data dependence between vertices $B_1$ and $B_2$. That is, there exists at least one variable v in $B_2$ whose value is computed in $B_1$. This type of variable (such as v) is called as computation variables in $B_1$ and is also known as reference variables in $B_2$, i.e., $B_1$ is $B_2$'s parent.*

Considering the circuit model in Figure 1, we can easily build *SDG* of the 17 sorted blocks of the equation system when performing topological sorting of the equation system into BLT form—all the nodes in one strongly connected component construct a block. As Figure 5a shows, a solving dependency graph is a directed graph without loops.

Once the *SDG* of an equation system is built, we can easily obtain the computing sequences through erasing the parent blocks [9], as shown in Figure 5b. Blocks in the same rectangle can be resolved in a random order.

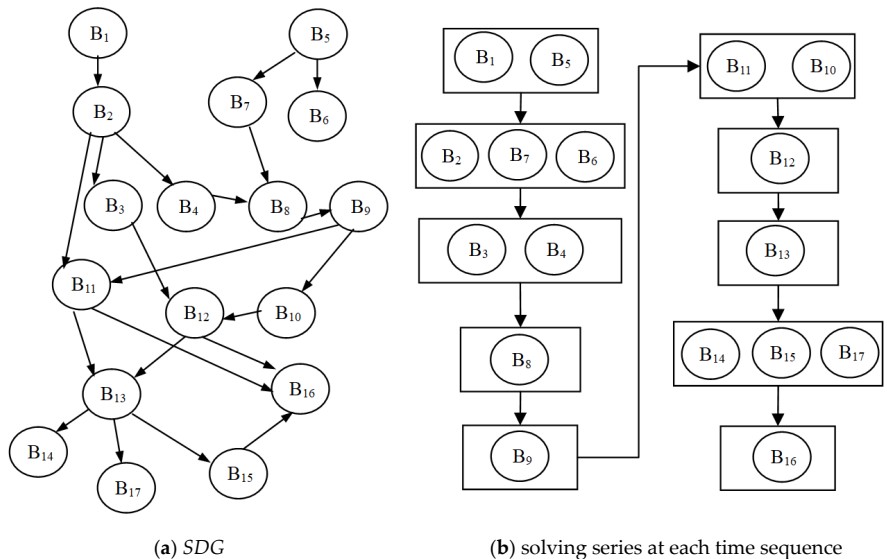

(**a**) *SDG*          (**b**) solving series at each time sequence

**Figure 5.** Solving dependency graph (*SDG*) and its solving series.

*4.2. Minimum Solving Graph*

Once the optimization model is built, we can construct its minimum solving graph based on the *SDG*. This method is based on the idea that if all the variables are resolved at first, we just need to resolve the blocks which are related to the optimization model. During the optimization iteration process, the simulation objective is to evaluate the values of objective functions and constraints according to the tuners and case parameters. Thus, the minimum solving graph is the sub-graph of *SDG* from the blocks of the tuners and case parameters to the blocks of constraints and objective functions.

The following are the three steps required to construct the minimum solving graph: getting the sub-graph coving post-blocks of parameters (including tuners and case parameters) in which all the blocks are affected by these parameters; getting the sub-graph coving pre-blocks of constraints or objective functions in which all these blocks affect blocks of objective functions or constraints; getting the intersection of the two sub-graphs. This intersection sub-graph is the very minimum solving graph. Given $G_P$ is the post-blocks of parameter *par1, par2, par3, . . .* ; $G_O$ is the pre-blocks of objects *obj1, obj2, obj3*; and $G_C$ is the pre-blocks of constraints *con1, con2, con3, . . .* ; then the minimum solving process is shown in Figure 6.

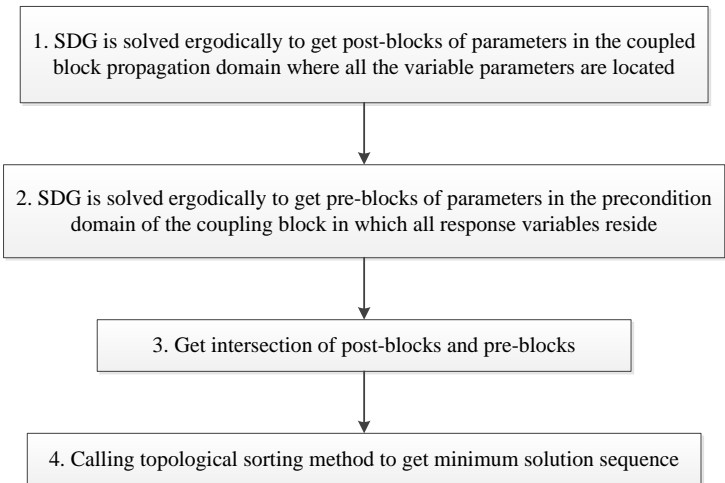

**Figure 6.** The minimum solving process on the minimum solving graph.

The specific algorithm implemented can be stated as follows:

$$MSG = G_P \cap (G_O \cup G_C)$$

The following Algorithm 1 illustrates the procedure of getting post-blocks of parameters:

---

**Algorithm 1. GetParPostBlocks**

---

*Input: s_strPar – a set of parameters*
        *SDG – solving dependency graph*
*Output: s_pBlks – set of post-blocks selected*
*for each block b in SDG do begin*
    *if intersection of b.s_Par and s_strPar is null or b is in s_pBlks then continue;*
    *clear set s;*
    *s = GetSuccesor (b, SDG, s_pBlks); //get successors of b in SDG*
    *Add s to s_pBlks; //select s*
*end for;*

---

In order to improve the efficiency of the algorithm, the *GetSuccessor* function (Algorithm 2) is passed with the set of blocks *s_pBlks* which have been selected:

---

**Algorithm 2. GetSuccesor**

---

*Input: b – the block*
        *SDG – solving dependency graph*
        *s_pBlks – set of blocks which has been selected*
*Output: s_pSucBlks – set of blocks selected*
*clear set s;*
*s = b.s_children;*
*for each block b1 in s do begin*
    *if b1 is in s_pBlks then continue;*
    *clear set s1;*
    *GetSuccesor ( b, SDG, s_pBlks); //call itself recursively*
    *Add b1 to s_ pSucBlks;*
*end for;*

---

Similarly, the following Algorithm 3 illustrates the procedure of getting pre-blocks of variables:

---

**Algorithm 3. GetVarPreBlocks**

   **Input**: *s_strVar – a set of Variables*
         *SDG – solving dependency graph*
   **Output**: *s_pBlks – set of blocks selected*
   **for each** *block b in SDG* **do begin**
     **if** *intersection of b.s_Var and s_strVar is null* **or** *b is in s_pBlks* **then continue;**
     *clear set s;*
     *s =* **GetProccesor***(b, SDG, s_pBlks); //Get Processors of b in SDG*
     *Add s to s_pBlks; //select s*
   **end for;**

---

Like the function *GetSuccessor*, *Getsuccessor* function (Algorithm 4) is also passed with the set of blocks *s_pBlks*:

---

**Algorithm 4. GetProccesor**

---

   **Input**: *b – the block*
         *SDG – solving dependency graph*
         *s_pBlks – set of blocks which has been determined*
   **Output**: *s_pProBlks – set of blocks*
   **clear set s;**
   **s = b.s_parents;**
   **for each** *block b1 in s* **do begin**
     **if** *b1 is in s_pBlks* **then continue;**
     *clear set s1;*
     **GetProccesor** *( b, SDG, s_pBlks); //call itself recursively*
     *Add b1 to s_ pProBlks;*
   **end for;**

---

The following Algorithm 5 illustrates the procedure of getting an intersection of two sets of blocks:

---

**Algorithm 5. GetMSGBlocks //getting set of blocks in a MSG**

---

   **Input**: *s_pParBlks—set of post- blocks of parameters*
         *s_pVarBlks—set of pre-blocks of output variables appearing in the objective functions or constraints functions*
   **Output**: *s_pMSGBlks—set of blocks in a MSG*
   **for each** *block b in s_pParBlks* **do begin**
     **if** *b is in s_pParBlks* **then**
       *add b to s_pMSGBlks;*
   **end for**

---

Algorithm 5 only gets the set of blocks in a MSG, the next algorithm, Algorithm 6, gets the series of sorted blocks:

---

**Algorithm 6. GetSortedBlocks**

---

   **Input**: *s_pMSGBlks – set of blocks in a MSG*
   **Output**: *v_ pMSGBlks – vector of sets of blocks which are sorted*
   **while** *s_pMSGBlks in not null* **do begin**
     **for each** *block b in s_pMSGBlks* **do begin**
       *clear set s;*
       **if** *intersection of b.s_pParents and s_pMSGBlks is null* **then**
         *add b to s;*
   **end for;**

---

Up until now, all the algorithms to build a partial simulation program based on an MSG have been narrated. The main procedure of generating a series of blocks of an MSG is illustrated as the following algorithm, Algorithm 7:

---

**Algorithm 7: GenerateMSGSortedBlocks**

---

　　//**generating minimum solving series of blocks in MSG**
*Input* SDG, s_pars, s_Vars
*Output* v_setMSGBlks
*Step1: s_pBlk1 = **GetParPostBlocks** (SDG, s_pars);*
*Step2: s_pBlk2 = **GetVarPreBlocks** (SDG, s_Vars);*
*Step3: s_pMSGBlks = **GetMSGBlocks** (s_pBlk1, s_pBlk2);*
*Step4: v_pMSGBlks = **GetSortedBlocks** (s_pMSGBlks);*

---

Using the algorithm above, we can build the minimum solving graph of the circuit model of example 1 as Figure 7 shows. Figure 7a–d illustrates Algorithms 1–4, respectively. The results of the sorted blocks are {$B_{11}$ | $B_{12}$, $B_{13}$, $B_{17}$}.

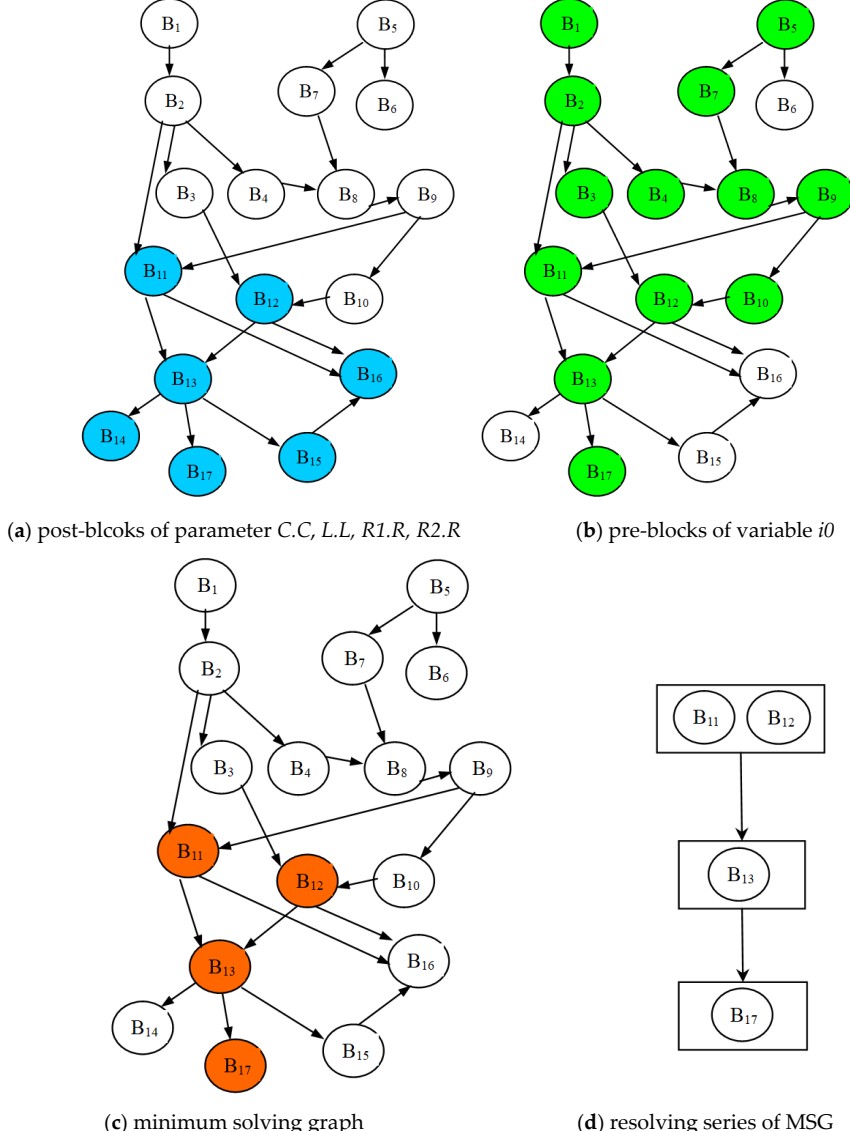

(**a**) post-blcoks of parameter *C.C, L.L, R1.R, R2.R*　　　　(**b**) pre-blocks of variable *i0*

(**c**) minimum solving graph　　　　　　　　(**d**) resolving series of MSG

**Figure 7.** Process of generating a MSG.

### 4.3. Optimization Iteration Process Based on Minimum Simulation

The minimum simulation program now can be built from the series of blocks of the MSG. When generating a minimum simulation code, the full simulation results resident in memory must be used to resolve the variables whose values changed in the MSG. All the values of unvaried variables are obtained at first through dynamics data exchange. In order to ensure all the values of all these variables at discrete time sequences have been saved, simulation steps must be matched between the partial resolving and full simulation. Thus, we adopt a constant steps (or the same varied steps) strategy in both the full simulation and minimum simulation code by setting the output interval to "interval length".

Based on the minimum simulation program, the process of parameter optimization has been improved, as Figure 8 shows. From the simulation model and the optimization model, the MSG is built at first. Then, full simulation is called only once to generate the full results in a mat file. According to the MSG and the full results, the minimum simulation program is finally built. Then, a simulation before evaluating the values of constraints and objective functions only needs to execute the minimum simulation program.

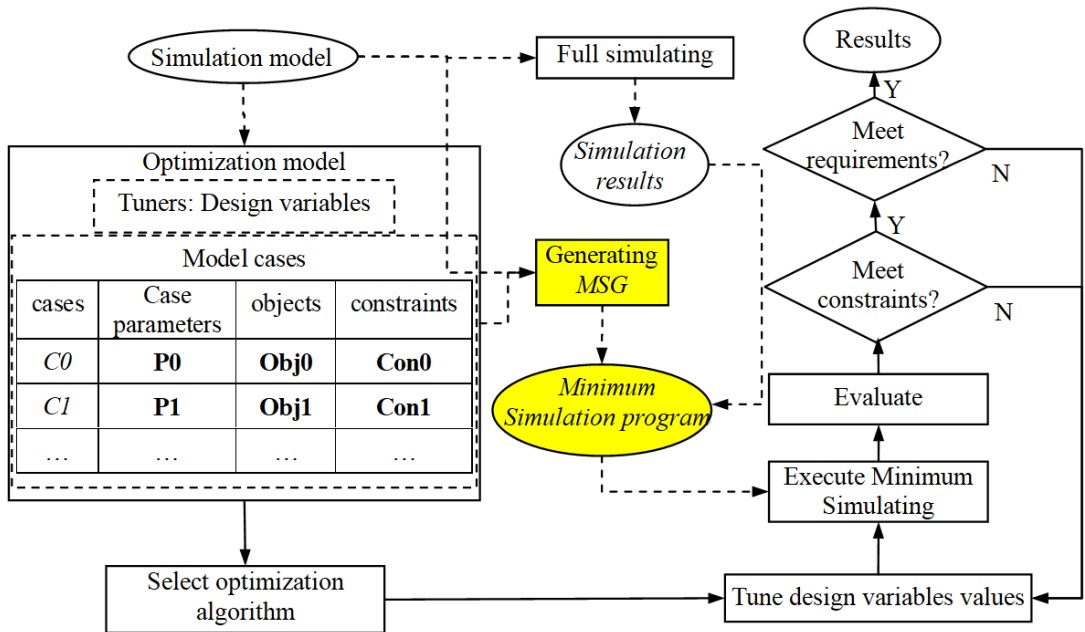

**Figure 8.** Optimization iteration process based on minimum simulation.

As mentioned above, the MSG is built not only by design parameters but also case parameters. As for the multi-case optimization model, another method of partial resolving is to build several different full simulation programs (the number of which equals the number of cases) according to different sets of values of the case parameters firstly, then constructing the minimum solving graph based only on the tuners (excluding case parameters), and lastly building several minimum simulation programs (number of which also equals to the number of cases). Thus, when evaluating the values of design functions of multi-cases, the multi minimum simulation programs will be called, respectively. The more efficient this method is, the greater the number of pre-blocks of case parameters and of optimization iterations is.

As for the optimization problem of the circuit example, we can get the optimal result, {$C.C$, $L.L$} = {0.033F, 0.0004H}, with the curves of the $i0$ ideal current and simulation response being shown in Figure 9. Using partial resolving during the optimization process can save time at about 10% when contrasted with full simulation. Actually, the different optimization models have different MSG. If we only select C.C as the tuners, then the MSG has only three blocks: $B_{11}$, $B_{13}$, and $B_{17}$. Accordingly, the

consuming time of the optimization could be reduced by 45%. However, if we select the parameter VS.signalSource.p_height as tuners, its MSG then covers nearly all the blocks, except four blocks: $B_1$, $B_2$, $B_3$, and $B_4$. This means time of partial resolving is close to that of the full simulation.

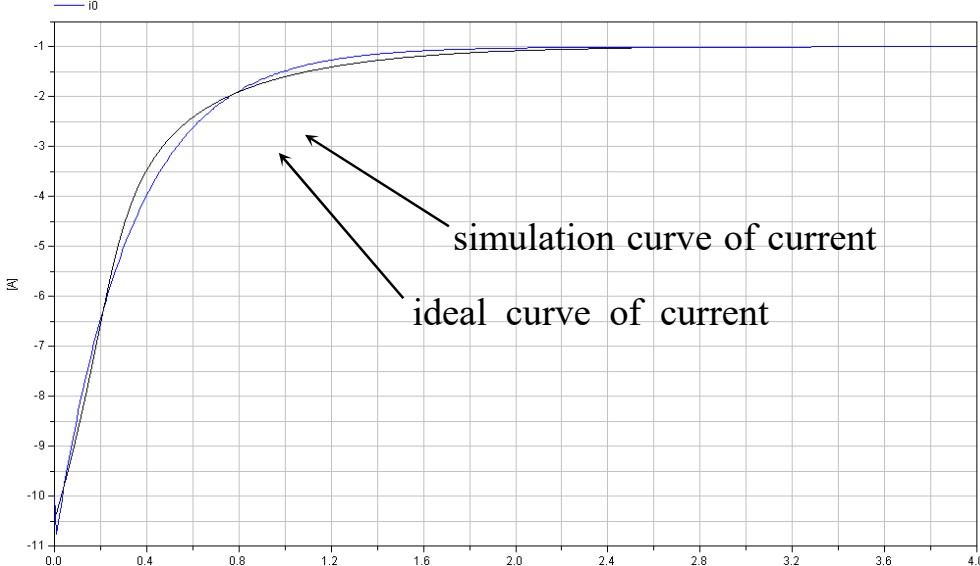

**Figure 9.** Optimal result of the current *i0* in the circuit of Figure 1.

## 5. Application

In order to save energy effectively, the economy of fuel is now given more and more consideration when designing vehicles, especially heavy trucks. Collaborating with Chinese Heavy Truck Corp., our group has built a *Modelica* library—InteDrive—to facilitate modeling and simulating for a heavy truck model. InteDrive includes the following sub-libraries: driveCycle, Driver (including BrakeControl, ThrottleControl, ShiftStrategy), engine, clutch, gearbox, brake, tire, vehicle, airbag, and fuel consumption. Based on the InterDrive library, a sample model of heavy truck—*veh_smcar*—has been built under MWorks, as Figure 10a shows.

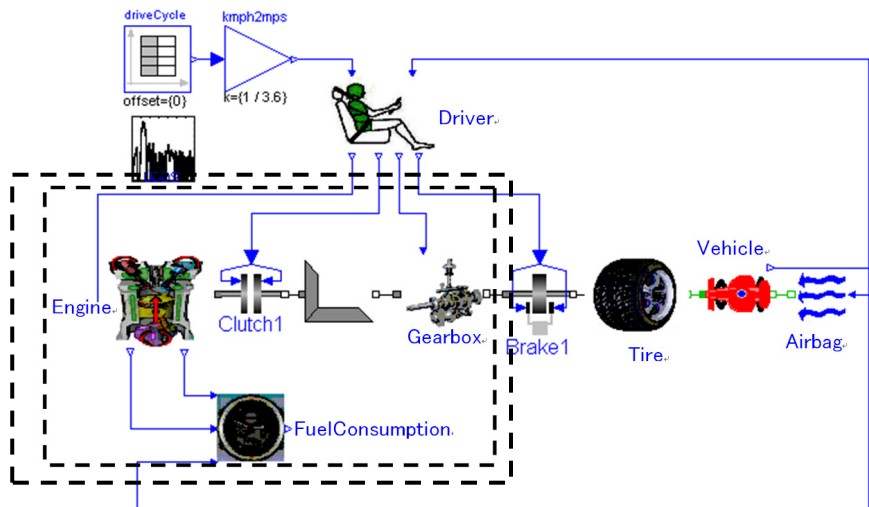

(**a**) *Modelica* model of *veh_smcar*

**Figure 10.** *Cont.*

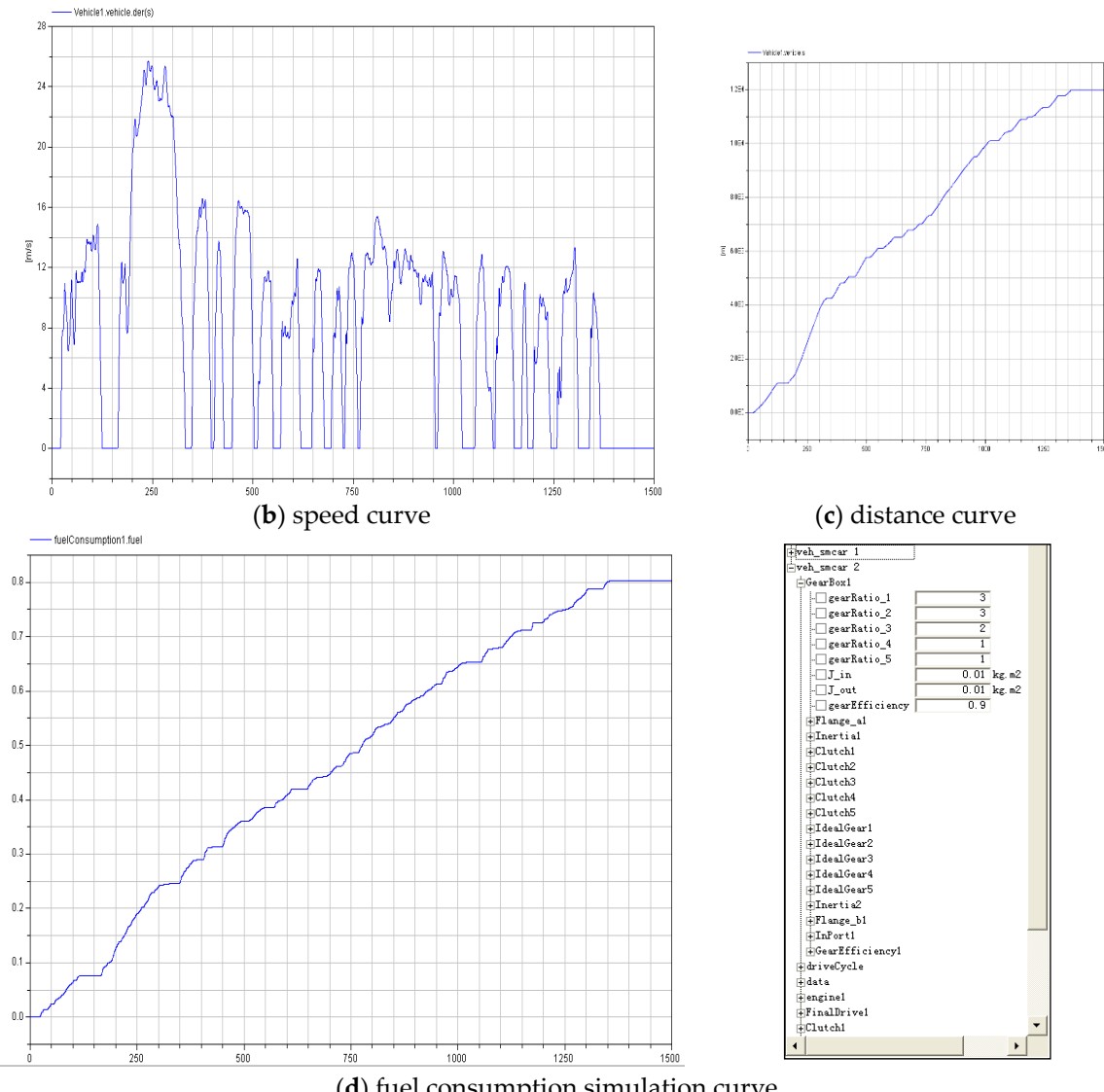

(**b**) speed curve

(**c**) distance curve

(**d**) fuel consumption simulation curve

**Figure 10.** *veh_smcar* (a sample of a heavy truck) model and simulation in 1500 s.

Under the MWorks platform, some main simulation curves can be obtained by setting the parameters: *GearBox1.gearRatio_1* = 3, *GearBox1.gearRatio_2* = 3, *GearBox1.gearRatio_3* = 2, *GearBox1.gearRatio_4* = 1, and *GearBox1.gearRatio_5* = 1, and adopting the predefined shift strategy of the component *Driver*:

```
parameter Real upshift[:] = {25, 50, 70, 100} "upshift table,(km/h)";
parameter Real downshift[:] = {20, 40, 60, 90} "downshift table,(km/h)";
Modelica.Blocks.Interfaces.InPort vehicle_speed;
Modelica.Blocks.Interfaces.OutPort requested_gear_number;
Integer current_gear(start = 1);
Integer speeds = size(upshift,1) + 1;
Real next_upshift;
Real next_downshift;
initial~algorithm
current_gear = 1;
next_upshift = if current_gear < speeds then upshift[current_gear] else Modelica.Constants.inf;
next_downshift = if current_gear > 1 then downshift[current_gear - 1] else -Modelica.Constants.inf;
```

```
algorithm
  when vehicle_speed.signal[1] > next_upshift then
    current_gear : = current_gear+1;
    next_upshift : = if current_gear<speeds then upshift[current_gear] else Modelica.Constants.inf;
    next_downshift : = if current_gear>1 then downshift[current_gear-1] else -Modelica.Constants.inf;
  end when;
  when vehicle_speed.signal [1] < next_downshift then
    current_gear : = current_gear-1;
    next_upshift : = if current_gear<speeds then upshift[current_gear] else Modelica.Constants.inf;
    next_downshift : = if current_gear>1 then downshift[current_gear-1] else -Modelica.Constants.inf;
  end when;
```

After simulation, the curves of vehicle speed, distance, and fuel consumption in 1500s are shown in Figure 10b–d, respectively. Total fuel consumption of *veh_smcar* in 1500 s was 8.1 L.

Next, we set up the optimization model:

```
find {gearRatio_1, gearRatio_2, gearRatio_3, gearRatio_4, gearRatio_5}
min: fuelComsumption1.fuel
s.t. max (vehicle1.speed) < Vj, j = 1,2,3
cases: three common shift strategies with different sets of table values of upshift and downshift.
    {upshift1, upshift2, upshift3} = {{25, 50, 70, 100}, {30, 55, 80, 110}, {20, 45, 65, 95}}
    {downshift1, downshift2, downshift3} = {{20, 40, 60, 90}, {25, 50, 70, 100}, {15, 40, 60, 90}}
    {V1, V2, V3} = {90 km/h, 100 km/h, 110 km/h}
```

After optimizing, the optimal values of tuners are as follows: *GearBox1.gearRatio_1* = 3.57, *GearBox1.gearRatio_2* = 2.01, *GearBox1.gearRatio_3* = 1.33, *GearBox1.gearRatio_4* = 1.00, and *GearBox1.gearRatio_5* = 0.75. The total fuel consumption was about 6.8 L. The fuel simulation curve in 1500 s is shown in Figure 11.

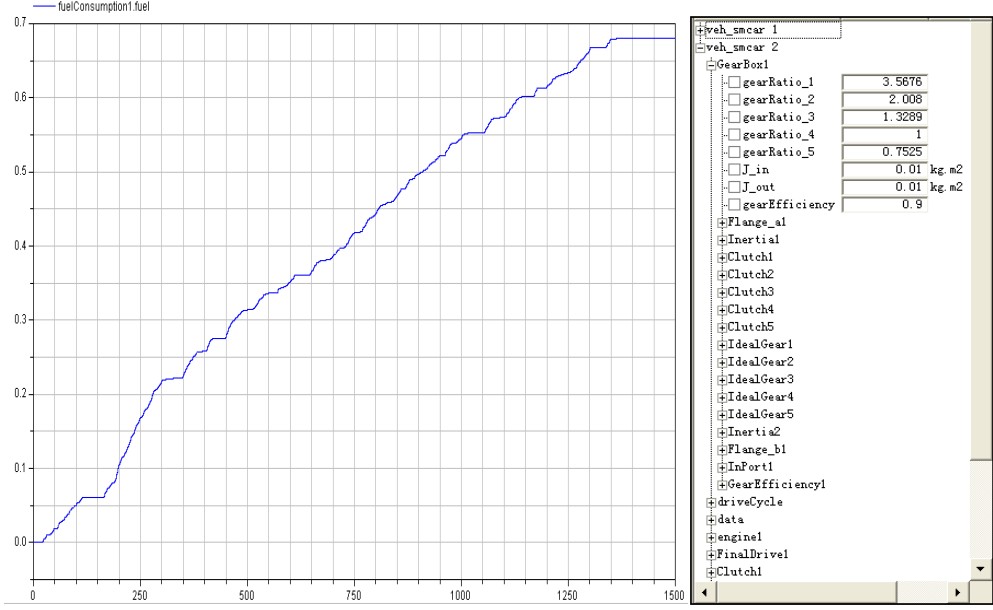

**Figure 11.** Fuel consumption curve after optimization.

By using the SQP algorithm, there are 12 iterations during the optimizing process. If simulation time is set up from 0 to 1500 s, the consuming time for full simulation is about 9.8 s under MWorks. After generating a minimum solving graph according to the optimization model, each partial simulation

time is about 5.6 s. Therefore, more than one third of time taken can be saved during the optimization process. As Figure 9a shows, only the variables in the dashed rectangle should be resolved when executing the partial simulation.

As for the *veh_smcar* model, if we select parameters in the component Engine (like *fc_inertia*) as tuners, its MSG will decrease greatly, while if we select parameters in the component driveCycle, such as the matrix table, then its MSG covers all the solving blocks.

## 6. Conclusions

Parameter optimization for a *Modelica* model needs to execute simulation repeatedly in order to evaluate design functions, and the efficiency of the process of design optimization depends on the efficiency of repetitious simulation to a great extent. Although there are many studies focusing on solving strategies of simulation, few of them pay attention to the efficiency of repetitious simulation after parameter tuning. Because only the equation blocks from design parameters to design functions of an optimization model need to be resolved during iterations, a minimum solving graph and partial resolving strategy based on a simulation model and optimization model is presented in this paper. By using this method, the more efficient it could be, the smaller the size of the MSG. Also, it would be useful for a model experiment with parameter tuning or parameter estimation.

For next study of this work, we are trying to generate several full simulation codes, corresponding MSGs and minimum simulation programs before optimizing. When executing partial simulation, the main optimization program calls one suitable minimum simulation program according to the values of parameters. Furthermore, a multiscale filtering algorithm and concept of network robustness [31] in the multiscale coordination control problem will be introduced to effectively guide us to enhance the efficiency and performance of the partial simulation resolving algorithm to a certain extent.

**Author Contributions:** Methodology, Y.L.; Software, K.H.; Validation, Y.L. and K.H.; Data curation, K.H.; Writing, Y.L. and K.H.; Visualization, K.H.

**Funding:** This research was fund by the National Natural Science Foundation of China. Discovery Grant NSFC-2018-51775472.

**Conflicts of Interest:** The authors declare no conflicts of interest.

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
