# Peer review of "Simulation Optimization for Complex Multi-Domain Physical Systems Based on Partial Resolving"

_processes, doi:10.3390/pr7060334_

Round 1

Reviewer 1 Report

This paper focuses on simulation optimization for complex multi-domain physical systems based on partial resolving. Parameter optimization for a Modelica model need to execute simulating repeatedly in order to evaluate design functions, and the efficiency of the process of design optimization depends on the efficiency of repetitious simulation to a great extent. Minimum solving graph and partial resolving strategy based on simulation model and optimization model is presented in this paper. As far as I am concerned, the results of this paper are meaningful. However, the presentation can be improved. There are some technical issues that need to be addressed. I will recommend this paper for publication only if all the comments below are addressed sufficiently.

(1) Line 19, "...,therefore this method could..." should be ", and therefore..."
(2) The citations in the text are wrong. They appear as i),ii) etc. It's interesting. How could this happen?
(3) Line 40, should "basis research" be "basic research"? Please double check. There seem to be many grammar errors and typos in the work.
(4) Line 47, "(DAE)" should not be in italic form.
(5) The contribution of the work should be further highlighted. How about its difficulty? For a paper to be published in Peocesses, a certain degree of novelty/difficulty is essential.
(6) Line 140, "the sorted 17 blocks is generated..." should be "are generated...". Moreover, the meaning of (B1) - B(17) should be explained. Please keep in mind most of the readers of Processes are not experts in this field.
(7) Some words on the top of Figure 4 tend to missing. Please re-generate this figure.
(8) The concept of dependency graph is interesting. In random graph theory, the same concept is intensively used. See the two seminal works: On the degree sequence of random geometric digraphs. This should be remarked to establish a new link to a wider audience.
(9) Algorithm 4 is not easy to follow. Some explanations are appreciated.
(10) The idea of sorted blocks are interesting, esp. Figures 5 & 6. This idea can potentially be used to structure insightful hierarchical structures. See the relevant work: Inhomogeneous long-range percolation on the hierarchical lattice. This should be briefly commented.
(11) The conclusion should be extended by adding some future directions and open problems. This would be very helpful for interested readers.
(12) A weakness of the work is that all elements involved in the system are assume to be normal. In reality, we have to consider faulty element. See the relevant work: Resilient multiscale coordination control against adversarial nodes. A remark on this aspect is needed.
(13) The reference list contains some inconsistencies. For example, in Ref. [8], it is shown as "Fritzson P" but in Ref. [9] it is shown as "J.W. Ding". Please pay attention to these small things. All of them ultimately contribute to the quality of the paper.

Author Response

Detailed Response to Reviewers 1

Authors: Kexi Hou, Yaohui Li

Title: Simulation Optimization for Complex Multi-domain Physical Systems based on Partial Resolving

Reviewer #1: This paper focuses on simulation optimization for complex multi-domain physical systems based on partial resolving. Parameter optimization for a Modelica model need to execute simulating repeatedly in order to evaluate design functions, and the efficiency of the process of design optimization depends on the efficiency of repetitious simulation to a great extent. Minimum solving graph and partial resolving strategy based on simulation model and optimization model is presented in this paper. As far as I am concerned, the results of this paper are meaningful. However, the presentation can be improved. There are some technical issues that need to be addressed. I will recommend this paper for publication only if all the comments below are addressed sufficiently.

(1) Line 19, "...,therefore this method could..." should be ", and therefore..."

Reply (1): Thanks for the valuable comments. ‘therefore’ has been replaced by ‘and therefore’.

(2) The citations in the text are wrong. They appear as i),ii) etc. It's interesting. How could this happen?
Reply (2): Thanks for the valuable comments. All references have been reordered. And exception characters have been deleted.

(3) Line 40, should "basis research" be "basic research"? Please double check. There seem to be many grammar errors and typos in the work.
(4) Line 47, "(DAE)" should not be in italic form.
Reply (3),(4): Thanks for the valuable comments. The modifications have been made. In addition, other grammatical errors in this article have also been corrected and marked in red.

(5) The contribution of the work should be further highlighted. How about its difficulty? For a paper to be published in Peocesses, a certain degree of novelty/difficulty is essential.
Reply (5): Thanks for the valuable comments. A new method named partial simulation resolving algorithm based on the set of input parameters and output variables for complex simulation model is proposed. A minimum solving graph of the simulation model is built according to the set of parameters, constraints and objects of the optimization model. In view of this, the simulation during the optimization iterative process needs only to resolve the variables on the minimum solving graph, which could decrease the simulating time greatly during every iterative step of the optimization process. Therefore, the proposed method is innovative to some extent. In addition, it takes quite a long time to construct and build the complex simulation model. For example, it takes us more than a year to build the heavy truck simulation model. In addition, the simulation process is also very time-consuming. It often takes about two hours for each numerical simulation. It has taken us nearly five years to complete this work. Therefore, it is difficult to add a new simulation example in a relatively short time.

(6) Line 140, "the sorted 17 blocks is generated..." should be "are generated...". Moreover, the meaning of (B1) - B(17) should be explained. Please keep in mind most of the readers of Processes are not experts in this field.

Reply (6): Thanks for the valuable comments. The specific meaning of each block from [B1] to [B17] can be found in Table 1.
(7) Some words on the top of Figure 4 tend to missing. Please re-generate this figure.
Reply (7): Thanks for the valuable comments. All the figures in this article have been re-generated.

(8) The concept of dependency graph is interesting. In random graph theory, the same concept is intensively used. See the two seminal works: On the degree sequence of random geometric digraphs. This should be remarked to establish a new link to a wider audience.
Reply (8): Thanks for the valuable comments. The sentence "The random geometric graphs [29] have been well studied and applied to alternative mechanisms wrapped in real-world complex networks." was added to this paper.

(9) Algorithm 4 is not easy to follow. Some explanations are appreciated.
Reply (9): The purpose of algorithm 4 is to obtain the solution blocks which are sorted. It generates the sorted blocks by adding and deleting ‘s’ in the loops. I think it is easy to follow. If any questions, please do not hesitate to contact me.

(10) The idea of sorted blocks are interesting, esp. Figures 5 & 6. This idea can potentially be used to structure insightful hierarchical structures. See the relevant work: Inhomogeneous long-range percolation on the hierarchical lattice. This should be briefly commented.
Reply (10): Thanks for the valuable comments. The sentence "In addition, the idea on the sorted blocks is interesting. It can potentially be used to structure insightful hierarchical structures. Yang [30] study a model for inhomogeneous long-range percolation on the hierarchical latticeof orderwith an ultrametric. The random geometric graphs and the idea on the sorted blocks are helpful for researchers to further understand the partial resolving algorithm." was added to this paper.

(11) The conclusion should be extended by adding some future directions and open problems. This would be very helpful for interested readers.
(12) A weakness of the work is that all elements involved in the system are assumed to be normal. In reality, we have to consider faulty element. See the relevant work: Resilient multiscale coordination control against adversarial nodes. A remark on this aspect is needed.
Reply (11): Thanks for the valuable comments. The remark “Further, a multiscale filtering algorithm and concept of network robustness [31] in the multiscale coordination control problem will be introduced to effectively guide us to enhance the efficiency and performance of the partial simulation resolving algorithm to a certain extent.” has been added into the conclusion.

Reply (12): Thanks for the valuable comments. The remark ”It is Noted that that all elements involved in the system are assumed to be normal. In addition, this method can only be used under the condition that equation system of the simulation model does not change when tuning the parameters. If the structure of the equation system changes, all the variables, solving dependency graph and the MSG could also change.” has been added into this paper.

(13) The reference list contains some inconsistencies. For example, in Ref. [8], it is shown as "Fritzson P" but in Ref. [9] it is shown as "J.W. Ding". Please pay attention to these small things. All of them ultimately contribute to the quality of the paper.

Reply (13): Thanks for the valuable comments. References have been revised.

Reviewer 2 Report

This paper proposes a method of reducing simulation optimization time for complex multi-domain physical systems by first constructing a minimum solving graph (MSG) and then performing the iterative simulation optimization steps using the MSG.

Although the goal of the paper is stated reasonably well in the Introduction, the main contribution of the paper, as described in Section 4, is difficult to follow due to a number of reasons, including poor English composition and grammar, and poor choice of words. Here are some suggested improvements:

Editorial Comments

1. Check English composition, spelling and grammar carefully from beginning to end. Also,

a) Replace the word “simulating” by “simulation” wherever appropriate, such as on lines 65, 76, 110, 171, etc.

b) Replace the word “estimating” by “estimation” on line 192, etc.

c) Replace the word “optimizing” by “optimization” on line 264, etc.

d) Replace the word “objects” by “objective functions” on line 250, etc.

Technical Comments

2. The citations need to be corrected, because they are indecipherable.

3. In Section 2.1, the goal of optimization is not clear. First explain in a few words what aspects/functions of the circuit you are trying to optimize. Also, before presenting the pseudocodes, explain in words how the above optimization goals are going to be achieved.

4. Rewrite Sections 3 and 4. In Section 3, first explain clearly how a multi-objective optimization is converted to a single objective function. If you are using “Weighted Sum Approach”, discuss its pros and cons because the solution obtained is not unique (depends on the choice of the weights).

Also, in Section 4, before presenting the pseudocodes, first describe clearly in words how a MSG for a given problem is constructed.

Author Response

Detailed Response to Reviewers

Authors: Kexi Hou, Yaohui Li

Title: Simulation Optimization for Complex Multi-domain Physical Systems based on Partial Resolving

Reviewer #2: This paper proposes a method of reducing simulation optimization time for complex multi-domain physical systems by first constructing a minimum solving graph (MSG) and then performing the iterative simulation optimization steps using the MSG.

Although the goal of the paper is stated reasonably well in the Introduction, the main contribution of the paper, as described in Section 4, is difficult to follow due to a number of reasons, including poor English composition and grammar, and poor choice of words. Here are some suggested improvements:

1. Check English composition, spelling and grammar carefully from beginning to end. Also,

a) Replace the word “simulating” by “simulation” wherever appropriate, such as on lines 65, 76, 110, 171, etc.

b) Replace the word “estimating” by “estimation” on line 192, etc.

c) Replace the word “optimizing” by “optimization” on line 264, etc.

d) Replace the word “objects” by “objective functions” on line 250, etc.

 Reply (1): Thanks for the valuable comments. Corresponding modifications have been performed.

Technical Comments

2. The citations need to be corrected, because they are indecipherable.

Reply (2): Thanks for the valuable comments. References have been corrected.

3. In Section 2.1, the goal of optimization is not clear. First explain in a few words what aspects/functions of the circuit you are trying to optimize. Also, before presenting the pseudocodes, explain in words how the above optimization goals are going to be achieved.

Reply (3): Thanks for the valuable comments. Table 1 have been added to explain the optimization goals.

4. Rewrite Sections 3 and 4. In Section 3, first explain clearly how a multi-objective optimization is converted to a single objective function. If you are using “Weighted Sum Approach”, discuss its pros and cons because the solution obtained is not unique (depends on the choice of the weights). Also, in Section 4, before presenting the pseudocodes, first describe clearly in words how a MSG for a given problem is constructed.

Reply (4): Thanks for the valuable comments.There are some basic background knowledge on Modelica optimization process. There is no need for further expansion here. But we add the sentence “A more specific process may be referred to in reference [9]. at the end of Section 3.3 so that the researchers who are interesting can found it. In addition, The minimum solving process on MSG (Figure 6) has been added into this paper to describe the process of MSG. The specific implement can be found from Algorithm 1 to Algorithm 4. For multi-objective optimization problems to be converted into single-objective optimization problems, it is not the focus of this paper. The purpose of this paper is to minimize the solution graph.

Round 2

Reviewer 1 Report

I appreciate the authors' remarkable turn around time. The paper has been substantially improved. I recommend acceptance after a couple of corrections. (1) Line 276, "Yang" should be "Shang". (2) Line 606, Ref. [29], the journal should be updated as "Applied Mathematical Sciences, 2010, 4(41): 2001--2012."

Author Response

I appreciate the authors' remarkable turn around time. The paper has been substantially improved. I recommend acceptance after a couple of corrections. (1) Line 276, "Yang" should be "Shang". (2) Line 606, Ref. [29], the journal should be updated as "Applied Mathematical Sciences, 2010, 4(41): 2001--2012."

Reply: Thanks for the valuable comments. Corresponding modifications have been made and highlighted in red.

Reviewer 2 Report

Revised version is acceptable for publication.

Author Response

Revised version is acceptable for publication.

Reply : Thank you very much, best wishes..